# Quality of care and post-discharge morbidity among children diagnosed with severe malaria in rural Uganda: A prospective cohort study

Jennifer M. Kniss[1], Georget Kibaba[2], Emmanuel Baguma[2], Sujata Bhattarai Chhetri[3], Cate Hendren[4], Moses Ntaro[5], Edgar Mulogo[5], Samson Karabyo[6], Ross M. Boyce[7] *

1 Department of Epidemiology, University of North Carolina at Chapel Hill, Chapel Hill, North Carolina, United States of America, 2 PHEALED, Bugoye, Uganda, 3 School of Medicine, University of North Carolina at Chapel Hill, Chapel Hill, North Carolina, United States of America, 4 Division of Medicine-Pediatrics, University of Massachusetts Chan Medical School, Worcester, Massachusetts, United States of America, 5 Department of Community Health, Mbarara University of Science and Technology, Mbarara, Uganda, 6 St. Paul's Level IV Health Center, Kasese, Uganda, 7 Institute of Global Health and Infectious Diseases, University of North Carolina at Chapel Hill, Chapel Hill, North Carolina, United States of America

* ross_boyce@med.unc.edu

**Data Availability Statement:** In accordance with institutional regulations, deidentified individual data that supports the results will be shared beginning 9

## Abstract

Pediatric severe malaria is a significant contributor of morbidity and mortality in Uganda. Most information is derived from tertiary referral centers and urban centers. Little is known about routine care or post-discharge outcomes in rural areas. We conducted a longitudinal cohort study of pediatric severe malaria at St. Paul's Level IV Health Center (SPHC) in Kasese, Uganda. We collected demographic, clinical, and laboratory results, and conducted follow-up 14 days post-discharge to assess patient outcomes in the immediate post-discharge period. The initial cohort included 187 children aged 0 to 17 years enrolled between July 9th, 2023 and January 9th, 2024. Almost all (94.7%) participants had a parasitological confirmed malaria diagnosis by rapid diagnostic tests or blood smear. While at SPHC, 95.7% of patients received 3+ doses of intravenous Artesunate, and 92.0% also received oral antimalarials. 62.0% had at least one symptom of severe malaria, with altered consciousness (40.6%) and convulsions (29.9%) the most frequently reported. 26.1% had evidence of severe malarial anemia (Hb <5 g/dl), of whom 93.5% received a blood transfusion. Most (82.2%) patients received care that we assessed as consistent with key elements of WHO management guidelines. We were able to contact 183 of the 187 patient caregivers post-discharge. Caregivers reported that 25.6% of patients were experiencing symptoms related to their hospitalization, with fever (18.5%) and nausea/ not feeding well (10.3%) reported most frequently. Children who experienced altered consciousness during their acute illness had 1.69 times the adjusted risk of reporting symptoms 14-days post-discharge compared to those who did not have altered consciousness (aRR: 1.69, 95% CI: 1.01–2.82). Six deaths were recorded, including three at SPHC and three post-transfer or discharge. Findings suggest that at private health facilities in rural areas, treatment appears to be consistent with

to 36 months following publication provided the investigator who proposes to use the data has approval from an Institutional Review Board (IRB), Independent Ethics Committee (IEC), or Research Ethics Board (REB), as applicable, and executes a data use/sharing agreement with UNC. The Principal Investigator should submit a request to the UNC Industry Contracting team (OSPContracting@unc.edu) to initiate the data use/sharing agreement.

**Funding:** Funding was provided by a Jefferson-Pilot Fellowship in Academic Medicine award from the UNC School of Medicine to RMB. The funders had no role in study design, data collection and analysis, decision to publish, or preparation of the manuscript.

**Competing interests:** The authors have declared that no competing interests exist.

guidelines. Future research should investigate high morbidity in the immediate post-discharge period.

## Introduction

According to the World Malaria Report, there were more than 600,000 malaria-related deaths in 2021 with the vast majority occurring among children under five years of age living in sub-Saharan Africa (SSA) [1]. Severe malaria (SM), typically categorized by various clinical syndromes such as severe anemia, cerebral malaria, and lactic acidosis, is responsible for the vast majority of these deaths [2]. While SM is almost always preventable with early diagnosis and treatment of uncomplicated disease, when present, SM is a medical emergency [3, 4]. Without timely and appropriate case management, ideally in a critical care setting, SM can quickly turn fatal [5]. Treatment of severe malaria with intravenous artesunate and supportive care has been shown in multiple clinical trials to save lives [6, 7]. Despite this, the global burden of SM remains stubbornly high, likely due to implementation gaps in existing evidence-based strategies. Previous studies suggest that improving overall adherence to WHO standards in health facilities, including treatment with artesunate IV and transfusion of blood products for severe anemia (hemoglobin < 5 g/dl), has been shown to reduce mortality from SM [8–10].

Identifying gaps in the quality of care and tracking clinical outcomes post-discharge for severe malaria patients is of utmost importance for reducing malaria-related morbidity and mortality. In Uganda, like many SSA countries, most studies of SM have been conducted at large, urban referral hospitals. Even at these facilities, the quality of SM case management is highly variable and often sub-optimal [11]. Less is known about the quality of care at lower-level facilities in rural areas, which account for the largest share of the population and are at the greatest risk of malaria [12, 13]. Understanding the quality of care received at lower-level facilities in rural areas is essential for guiding future efforts to improve case management practices in Uganda.

Additionally, evidence suggests that post-discharge outcomes are often poor for SM patients [14]. For example, 26.5% and 31.6% of children with cerebral malaria and severe malarial anemia respectively, compared to 14.5% of community-based controls, sought care in the 6 months period after discharge in Uganda [14]. However, evidence on patient outcomes post-discharge is limited, and few studies have assessed patient outcomes in the immediate post-discharge period. Given that many deaths occur outside of health facilities, this is a critical area of research [15]. Furthermore, few studies have assessed both the quality of care received while an inpatient, and patient outcomes in the immediate post-discharge period in the same population. Assessing outcomes post-discharge in addition to quality of care received can provide valuable insights into the impact of healthcare services on patients' well-being and quality of life.

Therefore, the overarching goal of this study was to evaluate the quality of care and post-discharge outcomes for pediatric patients admitted to a lower-level facility in rural western Uganda with severe malaria. To achieve this, we conducted a prospective, observational cohort study comparing routine care practices to current WHO guidelines for the management of SM. Key quality metrics included parasitologically confirmation of malaria, receiving at least three doses of intravenous artesunate followed by a prescription of ACTs, receiving hemoglobin (Hb) testing, and receiving a blood transfusion among patients with Hb<5 g/dL. Additionally, we followed patients longitudinally for a period of 14-days post-discharge to document interval care seeking and vital status.

## Methods

### Study site

We recruited participants from St. Paul's Level IV Health Center (SPHC) located in Kasese Township. This private, not-for-profit facility is led by the Anglican Diocese of Kasese and serves as the primary referral center for many of the surrounding rural areas. SPH is comprised of inpatient adult (general), pediatric, and maternity wards, a surgical unit, and an outpatient department (OPD) with differentiated services available for people living with HIV provided through a separate clinic. SPH provides services for malaria patients including malaria testing by RDT and light microscopy, intravenous artesunate, blood transfusion, oxygen via nasal passageways, and other supportive treatments. This level IV health facility is the first level where care is provided by physicians.

### Eligibility criteria

We recruited and enrolled children (age <18 years) admitted to either the pediatric or general wards being treated for malaria (either confirmed or suspected), who had an adult caregiver present to provide informed consent. We excluded outpatients, inpatients not being treated for malaria, and pregnant patients, as they would be admitted to a separate ward. For the purposes of this study, participants who tested positive for malaria on either a rapid diagnostic test or a blood smear were considered a parasitologically confirmed case.

### Data collection

Upon enrollment, study staff interviewed caregivers to collect demographic characteristics of the family and patients, including information on the age and sex of the patient, socioeconomic factors, and subdistrict of residence (S1 Appendix). Additionally, the study team asked caregivers about malaria preventative behaviors (e.g., bed net availability and use), care seeking behaviors for pediatric malaria and febrile illness, and history of hospitalization for malaria among any children in the household. Lastly, the study team collected information regarding the patient's present illness, including the number of days since symptom onset, and care-seeking prior to arrival at St. Paul's Health Center.

The study team reviewed each patient's chart daily. They abstracted physical exam findings noted upon admission, laboratory results including malaria testing, vital statistics, and medications received upon admission. On a daily basis, the clinician research assistants reviewed patient charts and recorded vital statistics (pulse, temperature, oxygen saturation), all medications and treatments received that day, and any laboratory results recorded that day. They also conducted and recorded daily physical exam findings each day. When information on patient charts were incomplete for temperature, pulse, oxygen saturation, or physical exam findings, the clinician research assistants performed these health assessments. Research assistants also recorded the medication prescribed upon discharge.

The study coordinator conducted interviews with the caregivers 14 days post-discharge by telephone. We asked questions about patient vital status, current symptoms, and additional care-seeking and medication usage post-discharge.

We obtained aggregated data on district level malaria cases derived from the HMIS system from the Kasese District Health Department.

### Categorization of symptoms and treatments

Participants were identified as having a fever either through caregiver reports (i.e., self-report) of the child exhibiting fever during the intake interview or by recording a temperature above

37.8 degrees Celsius at SPHC. We coded participants as having body aches, nausea/ poor feeding, headaches, and vomiting based on caregiver reporting during the intake interview. We recorded patients as having altered consciousness and/or convulsions if the caregiver reported these during the intake interview or if these symptoms were observed at SPHC. Participants were identified as having respiratory distress if they had an oxygen saturation reading of less than 90% at SPHC or if they received oxygen via nasal passageways at SPHC. We defined severe anemia as a hemoglobin of <5 grams per deciliter (g/dL) at any time during their inpatient stay at SPHC. The duration of inpatient stay was calculated by subtracting the date of admission from the date of leaving SPHC (regardless of through discharge, transfer, death, or leaving against medical advice). We assessed receipt of all medications and treatments, including artesunate, through review of the medical charts. All symptoms, care-seeking, and medications recorded at the 14-day follow-up were based on caregiver reporting.

## Statistical analysis

Data was collected in a RedCap database and analyzed using Stata version 18.0. We summarized participant characteristics of the total cohort and among children under five years and five years and above. We used the two-sample test of proportions to assess differences in symptoms between these age groups, with a p-value of less than 0.05 considered significant. We used the Wilcoxon rank-sum test to assess differences in the mean number of days between symptom onset and admission to SPHC. To assess associations between factors related with hospitalization and the presence of symptoms 14 days post-discharge, we ran a generalized linear model with a log link function and present crude and adjusted risk ratios (RRs). We included variables with a crude estimate p-value of less than 0.2 in the adjusted model. A p-value of less than 0.05 was considered statistically significant for the crude and adjusted models. Observations with missing data were excluded from analysis.

## Ethical review

Human subjects approval was obtained for this study from the Mbarara University of Science and Technology Research Ethics Committee (#2023–894) and University of North Carolina Institutional Review Board (#23–1013). Written informed consent was obtained from the caregiver of study participants.

## Results

### Sociodemographic and clinical characteristics

There were 187 eligible patients identified between July 9th, 2023, and January 9th, 2024, all of who agreed to participate. The median age of participants was five years (IQR 2–10). The vast majority (143, 77.7%) reported previous care seeking for the same illness, with most pursuing care at health facilities (76, 53.1%) or drug shops (61, 42.7%). Among those who sought care prior to admission, the majority (128, 69.2%) reported receiving some form of treatment, although under half (52, 40.6%) received an antimalarial (Table 1).

The median duration of symptoms prior to admission was 2 days (IQR 1–4) (**Table 1**). Older participants (ages ≥5 years) had a longer median duration of symptoms prior to admission (3 vs 2 days, p = 0.02) compared to younger participants. The most frequently documented symptoms were fever (177, 94.7%), nausea/poor feeding (152, 81.3%), headaches (141, 76.6%), and body aches (138, 75.0%). Children ≥5 years more frequently reported body aches (89.5% vs 59.6%; p<0.001) and headaches (89.6% vs 62.5%; p<0.001) compared to participants <5 years of age. The majority (116, 62.0%) of participants had experienced at least one

**Table 1. Characteristics of pediatric patients admitted with severe malaria.**

| Characteristic | Overall | Age <5 | Age 5–17 |
|---|---|---|---|
| | n = 187 | n = 91 | n = 96 |
| | No. (%) | No. (%) | No. (%) |
| **Patient Characteristics** | | | |
| Median age in years (IQR) | 5 (2–10) | - | - |
| Sex | | | |
| Male | 100 (53.5) | 51 (56.0) | 49 (51.0) |
| Female | 87 (46.5) | 40 (44.0) | 47 (49.0) |
| Household location | | | |
| Urban/ Peri-urban | 99 (52.9) | 49 (53.8) | 50 (52.1) |
| Rural | 88 (47.1) | 42 (46.2) | 46 (47.9) |
| Care seeking prior to admission[a] | | | |
| Yes | 143 (77.7) | 68 (75.6) | 75 (79.8) |
| No | | | |
| Missing | 3 | 1 | 2 |
| Site of care seeking | | | |
| Public/private health facility | 76 (53.1) | 30 (44.1) | 46 (61.3) |
| Drug shop | 61 (42.7) | 34 (50.0) | 27 (36.0) |
| Community Health Worker | 4 (2.8) | 3 (4.4) | 1 (1.3) |
| Other | 2 (1.4) | 1 (1.5) | 1 (1.3) |
| Received treatment or medication prior to admission | | | |
| Yes | 128 (69.2) | 61 (67.0) | 67 (71.3) |
| No | 57 (30.8) | 30 (33.0) | 27 (28.7) |
| Missing | 2 | 0 | 2 |
| Type of medication received (n = 128) | | | |
| First-line antimalarial medication | 52 (40.6) | 18 (29.5) | 34 (50.8) |
| Non-antimalarial only | 48 (37.5) | 25 (41.0) | 23 (34.3) |
| Ineffective antimalarial | 1 (0.8) | 0 (0) | 1 (1.5) |
| Unknown medication | 27 (21.1) | 18 (29.0) | 9 (13.4) |
| **Symptoms and Inpatient Stay** | | | |
| Symptoms onset, median days (IQR) | 2 (1–4) | 2 (1–3) | 3 (1–4.5) |
| Fever[b] | 177 (94.7) | 86 (94.5) | 91 (94.8) |
| Nausea/ poor feeding[b] | 152 (81.3) | 69 (75.8) | 83 (86.5) |
| Headache[b] | 141 (76.6) | 55 (62.5) | 86 (89.6) |
| Missing | 3 | 3 | 0 |
| Body ache[b] | 138 (75.0) | 53 (59.6) | 85 (89.5) |
| Missing | 3 | 2 | 1 |
| Vomiting[b] | 109 (58.6) | 51 (56.0) | 58 (61.0) |
| Missing | 1 | 0 | 1 |
| MUAC, median cm (IQR) | - | 13 (12, 14) | - |
| Any severe symptom | 116 (62.0) | 59 (64.8) | 57 (59.4) |
| Altered consciousness[b] | 76 (40.6) | 42 (46.2) | 34 (35.4) |
| Convulsions[b] | 56 (29.9) | 33 (36.3) | 23 (24.0) |
| Severe Anemia (Hb<5) | 46 (26.1) | 19 (22.6) | 27 (29.4) |
| Missing | 11 | 7 | 4 |
| Respiratory Distress | 26 (13.9) | 12 (13.2) | 14 (14.6) |
| Duration of inpatient stay, median nights (IQR) | 3 (2–5) | 3 (2–4) | 3.5 (2–5) |

(*Continued*)

**Table 1.** (Continued)

| Characteristic | Overall | Age <5 | Age 5–17 |
|---|---|---|---|
| | n = 187 | *n* = 91 | *n* = 96 |
| | No. (%) | No. (%) | No. (%) |
| Lost to Follow Up Post-Discharge | 4 (2.1) | 3 (3.2) | 1 (1.0) |

[a]Based on caregiver reporting only
[b]Based on either caregiver or clinical reporting

symptom consistent with a diagnosis of SM. Over a third (76, 40.6%) experienced changes in consciousness. More children <5 years experienced convulsions compared to those ≥5 years, (36.3% vs 24.0%, p = 0.07), although this difference did not achieve statistical significance. Over one in four patients (46, 26.1%) had severe anemia (Hb<5.0). Many patients (117, 64.6%) had a second diagnosis recorded at SPHC, with the most common including septicemia, anemia with no report of sickle cell disease (SCD), pneumonia, and SCD. However, we note that some of these diagnoses, such as anemia with no report of sickle cell disease, may have been caused by malaria. The median duration of inpatient stay was 3 nights (IQR: 2–5). Four patients were lost to follow-up after discharge from SPHC and were unable to be contacted for the 14-day follow-up call.

## Epidemiology of malaria in the Kasese District, Uganda

From July 1st to December 31st 2023, Kasese District reported a total of 58,848 suspected malaria cases among children ages 0–19 years old, among which 55,766 cases were confirmed. Notably, 12,380 confirmed cases (22.2%) occurred in children under the age of five. During this 6-month period, 191 children were admitted to St. Paul's Health Center due to malaria, of whom 183 (95.8%) were enrolled in the present study. Our investigation spanned two months of the dry season (July to August 2023) and four months of the rainy season (September to December 2023). The monthly district-level cases peaked during the rainy season in October (Fig 1). The greatest number of severe malaria cases admitted to SPHC peaked earlier, at the beginning of the rainy season in September.

## Treatment of severe malaria

Almost all participants (186 of 187, 99.5%) received an intravenous (IV) antimalarial medication. The vast majority of participants (179 of 187, 95.7%) received at least one dose of IV artesunate, while 19 (10.2%) received at least one dose of IV quinine. In 6 cases (3.2%), participants tested positive on a blood smear at least 24 hours after initial treatment with artesunate, and were then switched to IV quinine. Following intravenous treatment, 180 (96.3%) received a prescription for oral artemisinin-based combination therapy (ACTs) either while an inpatient at St. Paul's or upon discharge.

Participants also received a wide range of other medications and supportive care. The vast majority of patients received analgesics/antipyretics (181, 96.8%) and antibiotics (169, 90.4%) with amoxicillin being the most frequently prescribed. Over a third (67, 35.8%), including 43 (93.5%) of those with severe anemia, received a blood transfusion, 54 (28.9%) received haematinics or vitamins, 31 (16.6%) received another antiparasitic medication such as Albendazole, and 22 (11.8%) received supplemental oxygen via nasal cannula (Fig 2) at any point during admission.

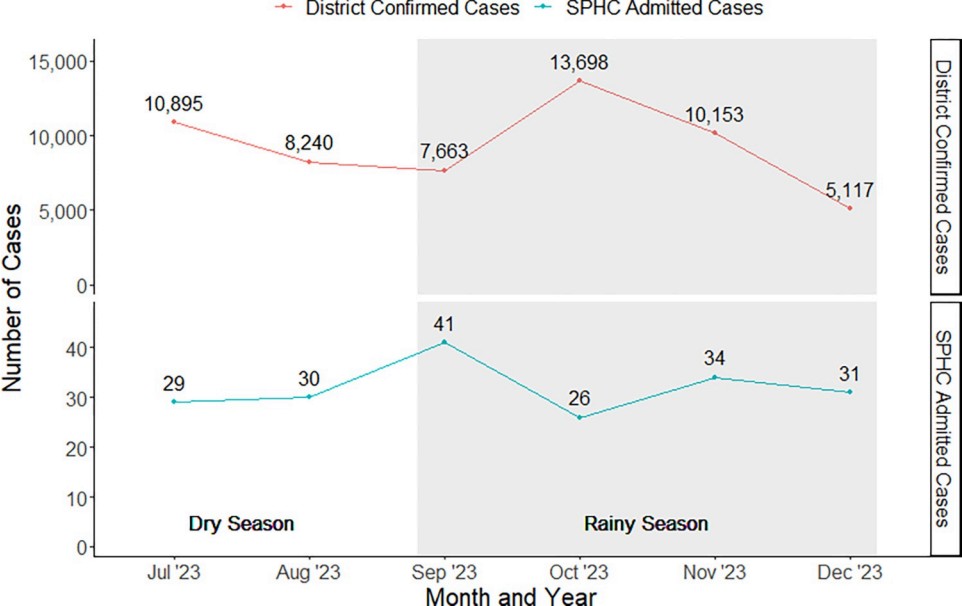

**Fig 1. Monthly malaria cases in Kasese District and at SPHC, July to December 2023.**

## Adherence to WHO malaria guidelines

All patients (187, 100%) received testing via blood smear, RDT, or both, and 177 (94.7%) had parasitologically confirmed malaria through these tests (Table 2). Blood smear was the most common diagnostic tool used, with all patients (187, 100%) receiving a blood smear at some point during their inpatient stay. Utilization of each diagnostic testing modality were similar between age groups. Almost all patients (178, 95.2%) received ≥ 3 doses of IV artesunate, including 97.8% of children under five years old. 93.0% of participants received both ≥3 doses of IV artesunate and a prescription for ACTs. Among 177 participants who were discharged from SPHC, excluding those who were transferred to another facility, left against medical advice, or died while at SPHC, two (1.1%) received 3+ doses of IV artesunate but did not

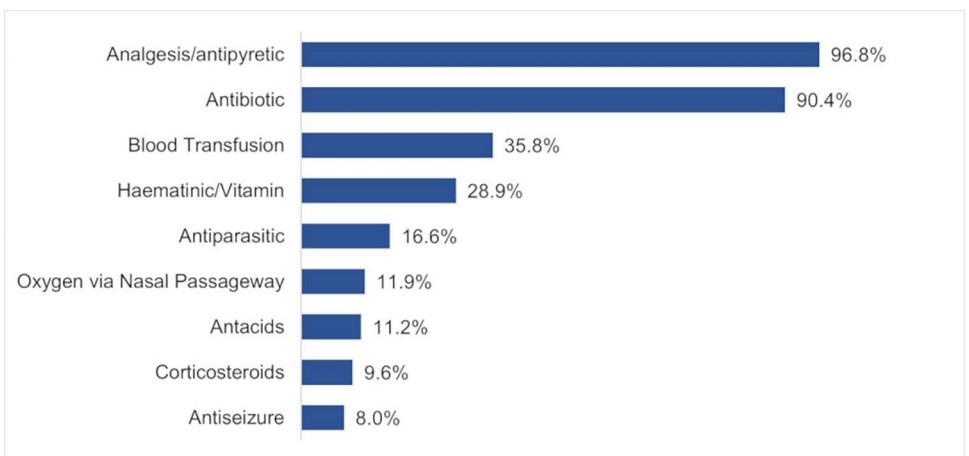

**Fig 2. Non-malarial medications and treatments prescribed at SPHC to 187 severe malaria patients.**

**Table 2. Impatient care and compliance with WHO guidelines for severe malaria.**

| | Overall | Age <5 | Age 5–17 |
|---|---|---|---|
| | n = 187 | n = 91 | n = 96 |
| | No. (%) | No. (%) | No. (%) |
| Parasitologically confirmed malaria, overall | 177 (94.7) | 87 (95.6) | 90 (93.8) |
| Parasitologically tested, overall | 187 (100.0) | 91 (100.0) | 96 (100.0) |
| Both blood smear and RDT | 140 (74.9) | 69 (75.8) | 71 (74.0) |
| Blood smear results only | 47 (25.1) | 22 (24.2) | 25 (26.0) |
| RDT results only | 0 (0) | 0 (0) | 0 (0) |
| No malaria test results | 0 (0) | 0 (0) | 0 (0) |
| Received 3+ Doses of IV Artesunate followed by Prescription of ACTs | | | |
| 3+ doses IV artesunate and ACT | 174 (93.0) | 86 (94.5) | 88 (92.6) |
| ACT, no 3+ doses IV artesunate | 8 (4.3) | 2 (2.2) | 6 (6.3) |
| 3+ doses artesunate, no ACTs | 4 (2.2) | 3 (3.3) | 1 (1.1) |
| Missing | 1 | 0 | 1 |
| Received Hb test | 176 (92.0) | 84 (88.3) | 92 (96.6) |
| Received blood transfusion among severe anemia patients (Hb<5) | 43 (93.5) | 18 (94.7) | 25 (92.6) |
| Antiepileptics received among children experiencing seizures at SPHC | 14 (25.0) | 9 (27.3) | 5 (21.7) |
| Diazepam | 7 (12.5) | 4 (12.1) | 3 (13.0) |
| Overall compliance[a] | 154 (82.4) | 78 (85.7) | 76 (79.2) |

[a]Blue rows indicate compliance to WHO guidelines. Treatment was considered compliant if patient had parasitologically confirmed malaria, received 3+ doses of IV artesunate followed by prescription of ACTs, received hemoglobin (Hb) testing, and received blood transfusion if hemoglobin was under 5

receive ACTs. Overall compliance with WHO Guidelines was estimated at 82.4%. This proportion was broadly comparable between children under age five (85.7%) and ages five and above (79.2%).

## Patient outcomes 14 days post- discharge

Six patients died (2.9%), including two patients under age five and four over age five (Table 3). Of these six patients, three died at SPHC, two died after transfer to another facility, and one was discharged home and died before the 14-day follow-up call. Deaths were reported on days 0, 2, and 7 post-admission at SPHC, and within 14 days post-discharge or transfer. At 14 days post-discharge, 172 (91.2%) participants were outpatients, while 5 (2.7%) were hospitalized. Overall, 45 (25.6%) caregivers reported that the child was still experiencing at least one symptom at the 14-day follow-up. The most common symptoms included subjective fever (18.5%) and nausea/not feeding well (10.3%, Table 3).

Nearly one in five (19.0%) participants reported additional care seeking after being discharged, with most seeking care at private/public health facilities (10.2%) and drug shops (7.0%, Table 3). A substantial proportion of participants (17.5%) reported that they were taking newly prescribed medication since being discharged, with 9 (5.1%) reporting newly prescribed antimalarial medication.

Children who experienced altered consciousness had 1.69 times the adjusted risk of experiencing symptoms post-discharge compared to those that did not have altered consciousness, when adjusted for care seeking prior to admission and respiratory distress (95% CI: 1.01–2.82; Table 4). We did not observe significant differences between children of different age groups or other factors.

**Table 3. Severe malaria patient outcomes 14 days post-discharge, n = 183[a].**

| | Overall | Age <5 | Age 5–17 |
|---|---|---|---|
| | n = 183[a] | n = 88 | n = 95 |
| | No. (%) | No. (%) | No. (%) |
| **Health Status** | | | |
| Vital Status | | | |
| Outpatient | 172 (91.2) | 85 (93.4) | 87 (90.6) |
| Deceased | 6 (3.3) | 2 (2.3) | 4 (4.2) |
| Hospitalized | 5 (2.7) | 1 (1.1) | 4 (4.2) |
| Health Returned to Normal[b] | 143 (80.8) | 72 (83.7) | 71 (78.0) |
| Currently experiencing symptoms | 45 (25.6) | 20 (23.3) | 25 (27.8) |
| Missing | 1 | 0 | 1 |
| Symptoms present | | | |
| Fever | 34 (18.5) | 18 (19.8) | 16 (17.2) |
| Nausea/ Not feeding well | 19 (10.3) | 5 (5.5) | 15 (15.1) |
| Cough | 16 (8.7) | 11 (12.1) | 5 (5.4) |
| Headache | 5 (2.7) | 2 (2.2) | 3 (3.2) |
| Lethargy | 4 (2.3) | 3 (3.5) | 1 (1.1) |
| Other[c] | 10 (5.4) | 6 (6.6) | 4 (4.2) |
| **Interval Care Seeking** | | | |
| Sought additional care | 35 (19.8) | 19 (22.1) | 16 (17.6) |
| Location of services | | | |
| Government/ private health center | 18 (10.2) | 9 (9.9) | 9 (9.9) |
| Drug Shop | 13 (7.0) | 6 (6.5) | 7 (7.3) |
| Traditional Healer | 4 (2.1) | 4 (4.4) | 0 (0) |
| **Medication Use** | | | |
| Currently taking medication    prescribed at St. Paul's | 8 (4.5) | 3 (3.5) | 5 (5.5) |
| Taking newly prescribed medications | 31 (17.5) | 15 (17.4) | 16 (17.6) |
| Newly prescribed antimalarial | 9 (5.1) | 6 (7.0) | 3 (3.3) |

[a]Total n = 183 is excluding four patients who were lost-to-follow up, including 3 under age five and 1 ages five and above

[b]Total denominator for this row and all rows below is n = 177, excluding the 6 patients who died

[c]Other includes vomiting (n = 2), stomach pain (n = 2), body ache (n = 1), altered mental status (n = 1), joint pain (n = 1), morning face swelling (n = 1), sore throat (n = 1), runny nose (n = 1)

## Discussion

In this prospective cohort study conducted at a level IV health facility that serves as one of the primary referral centers for severe malaria in the district, we observed that most (82.4%) patients received care that we assessed as consistent with key elements of the WHO management guidelines. Perhaps most importantly, the vast majority of patients received at least three doses of IV artesunate followed by a course of oral ACT (93.0%), both of which are substantially higher than recent studies conducted in similar SSA settings [11, 16, 17]. Despite the high level of care provided, a relatively high proportion of children reported persistent symptoms 14-day after discharge. This finding was unexpected and merits further investigation. Similarly, given that many patients had sought care prior to admission, additional efforts are required to ensure appropriate management at peripheral facilities to prevent treatment delays and subsequent development of severe malaria.

**Table 4. Crude and adjusted Risk Ratios (RR) for having symptoms at 14 days post-discharge, n = 177[a].**

| | n (%)[b] | Crude RR | 95% CI | Adjusted RR[c] | 95% CI |
|---|---|---|---|---|---|
| **Patient Characteristics** | | | | | |
| Age Group | | | | | |
| Under 5 | 20 (23.3) | 0.84 | 0.50–1.39 | | |
| Over 5 | 25 (27.8) | REF | - | | |
| Sex | | | | | |
| Male | 23 (24.2) | REF | - | | |
| Female | 22 (27.2) | 1.12 | 0.68–1.86 | | |
| Household location | | | | | |
| Urban/ Peri-urban | 23 (24.5) | 0.91 | 0.55–1.51 | | |
| Rural | 22 (26.8) | REF | - | | |
| **Care seeking prior to admission** | | | | | |
| Yes | 31 (23.5) | 0.69 | 0.41–1.16 | 0.66 | 0.39–1.10 |
| No | 14 (34.1) | REF | - | REF | - |
| Sought treatment prior to admission | | | | | |
| Public/ private health facility | 19 (26.0) | REF | - | | |
| Drug shop | 10 (18.5) | 0.73 | 0.37–1.44 | | |
| VHT | 2 (50.0) | 1.97 | 0.69–5.66 | | |
| Other | 1 (50.0) | 1.97 | 0.47–8.32 | | |
| Received medication prior to admission | | | | | |
| No | 15 (27.3) | REF | - | | |
| Yes | 29 (24.4) | 0.89 | 0.52–1.53 | | |
| Type of medication received | | | | | |
| Firstline anti-malarial    medication | 13 (27.1) | 1.20 | 0.64–2.26 | | |
| Other | 16 (22.5) | REF | - | | |
| **Symptom Onset and Inpatient Stay** | | | | | |
| Symptoms onset, days (IQR) | - | 0.96 | 0.85–1.08 | | |
| Duration of inpatient stay, nights (IQR) | - | 1.03 | 0.90–1.18 | | |
| **Severe Symptoms** | | | | | |
| Any severe symptom | 29 (27.1) | 1.17 | 0.69–1.99 | | |
| No severe symptom | 16 (23.2) | REF | - | | |
| Altered consciousness | 23 (33.8) | 1.66 | 1.01–2.74 | 1.69 | 1.01–2.82 |
| No altered consciousness | 22 (20.4) | REF | - | REF | - |
| Convulsions | 16 (32.0) | 1.39 | 0.83–2.33 | | |
| No convulsions | 29 (23.0) | REF | - | | |
| Severe Anemia (Hb<5) | 10 (23.8) | 0.89 | 0.48–1.65 | | |
| No severe anemia | 33 (26.6) | REF | - | | |
| Respiratory Distress | 8 (38.1) | 1.60 | 0.86–2.95 | 1.26 | 0.67–2.35 |
| No respiratory distress | 37 (23.9) | REF | - | REF | - |
| **Quality of Care** | | | | | |
| Overall compliance[d] | 37 (25.3) | 0.95 | 0.49–1.83 | | |
| Non-compliant | 8 (26.7) | REF | - | | |

[a]Total n ranges from 173 to 177 depending on missing values. Analysis excludes patients who died or were lost to follow-up

[b]Column presents the number and percentage of participants in each category who reported having symptoms 14-days post-discharge

[c]Adjusted risk ratio model included all variables with a crude estimate p-value of less than 0.2. This model included care seeking prior to admission (p = 0.16), altered consciousness (p = 0.05), and respiratory distress (p = 0.14)

[d]Tested positive for malaria, received 3+ doses of IV artesunate followed by a prescription of ACTs, received Hb test, and received blood transfusion among patients with Hb <5

Overall, the high level of adherence to management guidelines, including not just administration of artesunate, but also parasitological diagnosis (94.7%), hemoglobin testing (92.0%), and blood transfusion (93.5%), when indicated, is encouraging. These findings could be attributable to the PNFP structure of the clinic, which may facilitate higher staffing levels and compensation, while also minimizing stock-outs of key diagnostics and therapies. It's also possible that the study itself increased awareness of the importance of severe malaria care among providers (i.e., Hawthorne effect). Yet, if true, similar effects would be expected in other settings, which has not been reported. There were, however, still some areas for improvement. For example, one of the antiseizure medications used among patients was phenobarbital. The 2023 WHO Guidelines note that while phenobarbital reduces seizures, it was also shown to increase mortality in a double-blind, placebo-controlled evaluation [10]. Additionally, the WHO Guidelines do not recommend the use of high dose corticosteroids, and notes that use of corticosteroids increases the risk for certain complications [10]. Therefore, we recommend evaluation and revision of protocols for the use of these medications.

Notably, one in four children (25.6%) in our study were still reporting symptoms two weeks post-discharge, while almost one in five (19.8%) sought additional care during this same period. Even though few studies of severe malaria assess post-discharge morbidity and care-seeking, our findings appear higher than what has previously been reported [14, 18, 19]. For example, a rigorous study of post-discharge morbidity found 8% of severe malaria patients has sought care at five weeks after discharge [14]. There are multiple factors that could contribute to the high post-discharge morbidity and care-seeking. Given that fever was one of the most common indications for care seeking, it is possible that unrecognized or incident co-infections may be present. A prospective study of febrile illness conducted in Uganda from 2011–2013 found that 71.6% of participants with malaria had co-infections, with alphaviruses (e.g., chikungunya and O'nyong'nyong) and spotted fever rickettsiosis (SFR) being most common [20]. While there are no specific antiviral treatments against these alphaviruses, most patients in our cohort did not receive tetracycline antibiotics which would treat SFR. Additionally, some children may have experienced re-infection with malaria by the 14-day post-discharge follow-up. Our findings that 6 children continued to test positive after receiving artesunate and were then switched to quinine during hospitalization, along with emerging evidence from Uganda, also suggest the possibility of resistant parasites and recrudesce [21, 22]. Future research would be required to confirm this possibility. Given the potential for re-infection, implementing intermittent post-discharge malaria chemoprevention, as recommended in the 2023 WHO Malaria Guidelines, may reduce post-discharge morbidity and should be investigated further [10]. Receipt of IV artesunate may also be associated with adverse events such as abdominal pain and diarrhea, albeit infrequently, which may be contributing to a small proportion of the 14-day morbidity reported in this study [23].

Interestingly, we observed that children who experienced altered consciousness, a hallmark of cerebral malaria, during their acute illness had a higher adjusted odds of still experiencing symptoms 14 days post-discharge compared to children who did not experience altered consciousness (aRR: 1.69; 95% CI: 1.01, 2.82). One possible explanation for the increased risk of symptoms among pediatric severe malaria patients with altered consciousness may be post-infectious syndromes such as Post-malaria Neurological Syndrome (PMNS), defined as "the occurrence of de novo neurological signs after a symptom-free period following acute malaria associated with a negative blood smear and no retainable differential diagnosis" [24]. Although little is known about PMNS, it shares similarities with other central nervous system post-infectious syndromes that occur following recovery from certain viral or bacterial infections [24]. Symptoms reported in PMNS patients include fever, gastrointestinal symptoms, and more severe neurological symptoms such as mental confusion, seizures, cerebellar disorders, and

psychosis [24]. However, PMNS is thought to be very rare (although likely underreported), and has mostly been reported among adults. Furthermore, patients would require additional testing to rule out co-incident or new infections, including malaria and other potential infections, to be considered to have PMNS [24]. Future research should aim to assess the connection between cerebral malaria and post-discharge outcomes as well as additional risk factors for experiencing symptoms in the immediate post-discharge period for severe malaria patients.

Despite the relatively high post-discharge morbidity, mortality during hospitalization or the immediate post-discharge period was 2.9%. This is at the lower end of the risk of mortality reported in other studies in Africa, ranging from 2.2%-27% [16, 25]. The low mortality observed may be at least partly attributable to the high adherence to WHO treatment standards such as IV artesunate followed by ACT prescriptions, which have been found to reduce mortality rates [19]. However, it is possible that some participants in this study may have had a relatively severe disease other than malaria with unrelated malaria parasitemia. Therefore, symptoms among certain patients may not have actually been severe malaria, which may artificially lower the risk of mortality presented in this study depending on variances in eligibility criteria between studies.

We also observed a high prevalence of patients (77.7%) having sought care prior to admission at SPHC, mostly at lower level health facilities (53.1%) or drug shops (42.7%), which is consistent with findings elsewhere [26, 27]. Timely care seeking and early treatment, particularly within 24 hours of symptom onset, can prevent progression from uncomplicated to severe malaria [3]. Barriers to early treatment can include living far from health facilities providing appropriate care and financial barriers such as the cost of transportation to health facilities, and cycles of debt from previous hospitalizations [28]. Village Health Teams (VHTs) in Uganda VHTs may help to overcome these barriers to early malaria treatment, as they live closer to children and are trained to provide Integrated Community Case Management (iCCM) for illnesses such as malaria [29]. Additionally, VHTs have been shown reduce mortality among children [30]. Additionally, better training and resource availability at lower-level facilities to manage severe malaria cases, including basic amenity and equipment readiness, is important to reduce malaria mortality [31]. Ultimately, increased investment in quality-of-care improvements at lower levels of care may prevent progression of uncomplicated malaria to severe malaria and reduce morbidity and mortality observed at higher level facilities.

This study has a number of methodological strengths, including establishing temporality through prospectively following up patients over time, collecting history of illness early on during hospitalization to reduce recall bias, and collecting data on a wide variety of variables. However, it also has a number of important limitations, foremost of which was the observational design. The health center may have performed better due to knowing that the study team was observing severe malaria patients and the care provided. Additionally, study data was abstracted from clinical records, typically hand-written patient charts, which may have been incomplete or difficult to read contributing to measurement error. We attempted to minimize this by conducting data quality checks. Furthermore, while we considered ACT prescription upon discharge as compliant with WHO guidelines, we have no confirmation that these patients filled the prescription and took the medication. This could potentially influence 14-day follow-up findings. Additionally, data on history of illness, including questions about the specific day when the child started feeling ill and medications received before arriving at SPHC, were based on caregiver reporting, which could introduce recall bias. Moreover, certain caregiver-reported symptoms collected at the 14-day follow-up call were subjective, such as fever and not feeding well, which could introduce misclassification of the outcome. In addition, many patients had other diagnoses in addition to malaria. Therefore, the severity of

illness in some children may have been primarily caused by another serious illness with incidental parasitemia rather than by severe malaria. Lastly, overall compliance calculations did not include certain recommendations from WHO guidelines such as monitoring blood glucose levels and weighing patients upon admission to determine medication dosage, limiting comparability to certain other studies.

## Conclusions

This study fills an important gap in the literature by assessing both the quality of care compared to WHO standards and patient outcomes in the immediate post-discharge period. We found that treatment is generally compliant with WHO standards at a rural private level IV health center. The high prevalence of patients still experiencing symptoms and having sought additional care 14 days post-discharge warrants further investigation.

## Supporting information

**S1 Checklist. Inclusivity in global research.**
(DOCX)

**S1 Appendix. RedCap data collection surveys.**
(PDF)

## Acknowledgments

We appreciate the support of the staff at St. Paul's Level IV Health Center, for their access to facilities and equipment and unwavering support of the study. We thank the patients and their families at St. Paul's Level IV Health Center who participated in this study.

## Author Contributions

**Conceptualization:** Moses Ntaro, Edgar Mulogo, Ross M. Boyce.

**Formal analysis:** Jennifer M. Kniss, Ross M. Boyce.

**Funding acquisition:** Ross M. Boyce.

**Investigation:** Jennifer M. Kniss, Georget Kibaba.

**Methodology:** Emmanuel Baguma, Edgar Mulogo, Ross M. Boyce.

**Project administration:** Emmanuel Baguma, Ross M. Boyce.

**Resources:** Jennifer M. Kniss, Sujata Bhattarai Chhetri, Cate Hendren, Ross M. Boyce.

**Supervision:** Georget Kibaba, Emmanuel Baguma, Samson Karabyo.

**Writing – original draft:** Jennifer M. Kniss, Ross M. Boyce.

**Writing – review & editing:** Jennifer M. Kniss, Georget Kibaba, Emmanuel Baguma, Sujata Bhattarai Chhetri, Cate Hendren, Moses Ntaro, Edgar Mulogo, Samson Karabyo, Ross M. Boyce.

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
