## [Decision Letter · Decision Letter 0]

3 Jul 2024

PGPH-D-24-01221

Quality of Care and High Post-Discharge Morbidity Among Pediatric Severe Malaria Patients in a Rural Area of Uganda: A Prospective Cohort Study

Dear Dr. Boyce,

Thank you for submitting your manuscript to PLOS Global Public Health. After careful consideration, we feel that it has merit but does not fully meet PLOS Global Public Health’s publication criteria as it currently stands. Therefore, we invite you to submit a revised version of the manuscript that addresses the points raised during the review process.

The goals and potential impact of your work were appreciated by reviewers. A number of suggestions for improvement have been made and are summarized below, most of which are related to clarifications of the methodology and results, as well as additional aspects to incorporate into the discussion. We would welcome receiving your revised manuscript and hope that the reviewers' feedback will be constructive and helpful in the process. 

We look forward to receiving your revised manuscript.

Kind regards,

Amy Kristine Bei

Academic Editor

Journal Requirements:

Reviewer's Responses to Questions

**Comments to the Author**

1. Does this manuscript meet PLOS Global Public Health’s publication criteria? Is the manuscript technically sound, and do the data support the conclusions? The manuscript must describe methodologically and ethically rigorous research with conclusions that are appropriately drawn based on the data presented.

Reviewer #1: Yes

Reviewer #2: Yes

2. Has the statistical analysis been performed appropriately and rigorously?

Reviewer #1: Yes

Reviewer #2: Yes

3. Have the authors made all data underlying the findings in their manuscript fully available (please refer to the Data Availability Statement at the start of the manuscript PDF file)?

Reviewer #1: No

Reviewer #2: Yes

4. Is the manuscript presented in an intelligible fashion and written in standard English?

Reviewer #1: Yes

Reviewer #2: Yes

5. Review Comments to the Author

Reviewer #1: This is a pretty straight forward description of a study investigating the in-facility treatment and post-discharge outcomes of children presenting to a lower-level health centre in Uganda with features of severe malaria. The paper is generally well written and referenced and the statistics seem appropriate and nicely summarised in sensible tables.

Major comments

1) The title is a bit confusing. I was expecting to read a paper that was aimed at investigating the association between the quality of care received in the health facility and the post-discharge outcome. It soon became apparent that >80% of the children received treatment that complied with WHO standards and that the authors made no attempt to relate care-quality to outcome. I suggest a change to the title!

2) The post-discharge morbidities recorded were rather soft and subjective. Some (such as "not feeling well") could apply to any child who was not completely back to normal (which would seem unlikely in many children who had been admitted to hospital with a life-threatening condition only 2 weeks after discharge). I wonder how standard the methods and definitions are?

3) In the results section and in Figure 1, the authors present data on the number of patients with suspected or confirmed malaria in Kasese District between July and December 2023. However, these data don’t seem to be derived from this study and there is no description of the methods used in the methods section. If not part of the study then they should not be included in the results but referred to (with an appropriate citation) either in the introduction or discussion. They could potentially remain in the figure for comparative purposes with an appropriate foot note and reference.

4) The authors shouldould probably be more cautious in their interpretation of persisting parasitaemia beyond 24 hours. Although most will clear parasites within that time, this is not universal, even in the era before resistance (see for example White Malar J (2017) 16:88).

5) The authors could also be a bit more clear about the fact that this group of children probably included a significant number who didn't actually have severe malaria but were suffering instead from another serious illness with incidental parasitaemia (see for example Watson et al., Sci. Transl. Med. 14, eabn5040 (2022 and Watson, Ndila, et al. eLife 2021;10:e69698). They do mention it briefly in the discussion but I think they could be more upfront about it.

6) Incidentally, do the authors have any other data that would help them tease apart the previous point?

Minor comments

1) In the abstract, the authors state that there were 6 deaths recorded by the 14-day post-discharge follow-up. This is a bit deceptive because 3 of the deaths occurred in hospital and only 3 occurred post discharge. Please clarify in the abstract.

2) Reference 5 should be replaced with the original citation rather than a citation to the WHO guideline.

Reviewer #2: Overall: This was a well-written manuscript of an observational longitudinal cohort study evaluating care and adherence to WHO guidelines for pediatric severe malaria cases. The study took place at a private health facility in Kasese, Uganda during the end of the dry season into the rainy season. Participants were followed from admission to discharge and contacted 14 days post-discharge to assess for any symptom onset. This was a very interesting study and I appreciate the authors’ thorough description of the epidemiology of malaria in the district in addition to the findings from the cohort. I have a few comments for the author’s consideration.

Methods section

1. Eligibility criteria section - Could authors confirm whether the eligibility criteria included only those with severe or complicated malaria or did it include all children? Eligibility criteria seems like any confirmed case was eligible.

2. Statistical analysis: “We ran a generalized linear model with a log link function to assess crude and adjusted risk ratios (RRs) for the binomial outcome of experiencing symptoms 14 days post-discharge.” - Perhaps authors can clarify here what the research question was here (e.g., assess predictors of symptom onset post-discharge)?

Results:

1. Line 241 - Could authors specify what treatments the participants who did not receive 3+ doses of IV artesunate or ACT?

2. Table 1 - Regarding “Received treatment or medication prior to admission” and “Type of medication received” - Do authors mean treatment/medication to refer to any medications or specific to treatment of malaria?

3. Table 2 “Received 3+ Doses of IV Artesunate followed by Prescription of ACTs” sub-section - It might be more informative if authors could provide the n (%) for those who received 3+ dose IV only, and those who received ACT prescriptions only.

4. Table 2 - Authors may consider adding a footnote that blue indicates compliance in the “a” footnote.

5. Line 211 - “peaked” rather than “peeked”

6. Fig 1 - Could authors provide trends in severe malaria cases to reference their statement?

7. Line 265 - Medication for treatment of malaria or any medication?

8. Line 275-277 - Could authors also describe what proportion of those who experienced altered consciousness received IV artesunate + ACTs?

9. Table 4 - Could authors report p-values, only for the sake of understanding how authors decided which variables when into the adjusted model.

10. Table 4 - Authors could consider adding n (%) for the outcome in addition to the RRs to understand the proportions of those who experienced each symptom.

Discussion:

1. Line 291 - Wasn’t this 81.3% not 82.2% as per Table 2?

2. I think a major comment would be that authors could consider adding some discussion around the recent 2022 WHO guidelines regarding post-discharge malaria chemoprevention. One of the authors’ main points is that a high proportion of participants are experiencing symptoms mainly fever. While symptom classification was based on self-report, it is possible (though unknown) that children were re-infected or experienced recrudescence.

3. Line 294 - I think it’s important to report what proportion of children received IV artesunate only without a course of ACT. IV artesunate will rapidly reduce parasite burden, but it has a fairly short half-life and thus needs to be followed with ACTs to clear residual parasitemia. While I expect this # to be small, IV artesunate without follow up of oral ACT is concerning re: drug resistance and recrudescence.

4. Line 330 - Did these 6 children also receive ACTs following IV artesunate? And how long after were these children tested? I don’t see this in the results section, but maybe I missed it? This might be a good section where authors could discuss post-discharge malaria chemo prevention.

5. Limitations - RE: caregiver reporting - I think this is more generally a problem with misclassification of the outcome rather than recall bias. For example, fever is subjective.

---

## [Decision Letter · Decision Letter 1]

10 Sep 2024

PGPH-D-24-01221R1

Quality of Care and Post-Discharge Morbidity Among Pediatric Severe Malaria Patients in a Rural Area of Uganda: A Prospective Cohort Study

Dear Dr. Boyce,

Thank you for submitting your manuscript to PLOS Global Public Health. We provisionally accept your manuscript but ask that you address the small minor points raised by the reviewer during the last round of review. We invite you to submit a revised version of the manuscript that addresses the points raised during the review process.

The suggested changes to be incorporated are as follows:

"I thank the authors for adequately responded to my reviewer comments. I have a few additional minor requests for the authors, which I list below:

1. Table 4 - Please revise “RR” column to “Crude RR”

2. The study eligibility criteria was inpatient children of pediatric and general wards with confirmed or suspected malaria. Thus, this cohort of children included those with and without severe malaria diagnosis. In fact, authors state that only 62% of children had a symptom consistent with severe malaria (lines 191-192). I see that authors responded to Reviewer #1’s comment regarding this point (Major Comment 1.5), but I agree that in this revised version, authors should be more upfront about this point throughout the manuscript, including a change to the title which states this is a study of pediatric severe malaria cases (which is not necessarily the case). Moreover, authors could consider adding “severe malaria diagnosis” as a risk factor for patient outcomes 14-days post-discharge."

We look forward to receiving your revised manuscript.

Kind regards,

Amy Kristine Bei

Academic Editor

Journal Requirements:

Additional Editor Comments (if provided):

Reviewers' comments:

Reviewer's Responses to Questions

**Comments to the Author**

1. If the authors have adequately addressed your comments raised in a previous round of review and you feel that this manuscript is now acceptable for publication, you may indicate that here to bypass the “Comments to the Author” section, enter your conflict of interest statement in the “Confidential to Editor” section, and submit your "Accept" recommendation.

Reviewer #1: All comments have been addressed

Reviewer #2: All comments have been addressed

2. Does this manuscript meet PLOS Global Public Health’s publication criteria? Is the manuscript technically sound, and do the data support the conclusions? The manuscript must describe methodologically and ethically rigorous research with conclusions that are appropriately drawn based on the data presented.

Reviewer #1: Yes

Reviewer #2: Yes

3. Has the statistical analysis been performed appropriately and rigorously?

Reviewer #1: Yes

Reviewer #2: Yes

4. Have the authors made all data underlying the findings in their manuscript fully available (please refer to the Data Availability Statement at the start of the manuscript PDF file)?

Reviewer #1: Yes

Reviewer #2: Yes

5. Is the manuscript presented in an intelligible fashion and written in standard English?

Reviewer #1: Yes

Reviewer #2: Yes

6. Review Comments to the Author

Reviewer #1: All my comments have been addressed.

Reviewer #2: I thank the authors for adequately responded to my reviewer comments. I have a few additional minor requests for the authors, which I list below:

1. Table 4 - Please revise “RR” column to “Crude RR”

2. The study eligibility criteria was inpatient children of pediatric and general wards with confirmed or suspected malaria. Thus, this cohort of children included those with and without severe malaria diagnosis. In fact, authors state that only 62% of children had a symptom consistent with severe malaria (lines 191-192). I see that authors responded to Reviewer #1’s comment regarding this point (Major Comment 1.5), but I agree that in this revised version, authors should be more upfront about this point throughout the manuscript, including a change to the title which states this is a study of pediatric severe malaria cases (which is not necessarily the case). Moreover, authors could consider adding “severe malaria diagnosis” as a risk factor for patient outcomes 14-days post-discharge.

---

## [Editor Report · Decision Letter 2]

17 Sep 2024

Quality of Care and Post-Discharge Morbidity Among Pediatric Severe Malaria Patients in a Rural Area of Uganda: A Prospective Cohort Study

PGPH-D-24-01221R2

Dear Dr. Boyce,

We are pleased to inform you that your manuscript 'Quality of Care and Post-Discharge Morbidity Among Pediatric Severe Malaria Patients in a Rural Area of Uganda: A Prospective Cohort Study' has been provisionally accepted for publication in PLOS Global Public Health.

Thank you again for supporting Open Access publishing; we are looking forward to publishing your work in PLOS Global Public Health. Congratulations again!

Best regards,

Amy Kristine Bei

Academic Editor
